# Exaptation in Transitional Urban Morphologies: First Notes on the Dynamics of Urban Form Read through the Theories of Natural Evolution

Marco Trisciuoglio

"Transitional Morphologies" Joint Research Unit, Politecnico di Torino and Southeast University Nanjing, Nanjing 210018, China; marco.trisciuoglio@polito.it; Tel.: +39-011-0906528

**Abstract:** Studying the dynamics of urban form means questioning the processes of evolution of the form in general. The current discussion on the architecture of buildings and urban spaces has drawn the concept of adaptation from theories of natural evolution. These notes propose a reflection on the opposite and controverse concept of exaptation as it was proposed by the biologist and paleontologist Stephen Jay Gould in 1982. Through some examples (the different transformations of some Roman amphitheaters of the imperial age and the metamorphoses that occurred in the 20th century to some Chinese urban fabrics, originally made by courtyard houses), it is possible to extend to urban forms the idea of the casual co-optation for new uses of organs and anatomical parts developed for other reasons. This kind of reflection opens up innovative considerations on the potential of transitional urban analysis and its repercussions on evolutive urban transformation processes.

**Keywords:** transitional morphologies; dynamics of urban forms; exaptation/adaptation

## 1. Introduction

This paper delves into the notion of exaptation within the context of urban morphology and the dynamics governing urban form. It expounds upon how preexisting structures have undergone co-optation and transformation to accommodate novel functionalities over time. Even considering its controverse role in natural sciences, the concept of exaptation yields fresh perspectives on evolving urban morphologies and their potential for pioneering urban transformation processes. Moreover, the imperative need for more sophisticated conceptual and technical instruments within urban morphology is underscored here, in order to align with the swift metamorphosis of urban spaces. In essence, the concept of exaptation broadens the comprehension of adaptation in urban morphology, enabling a more comprehensive consideration of permanence, variability, and adaptability in urban design. Ultimately, these notes propose the incorporation of exaptation within the general theory of architecture, in the tradition of the Biological Analogy described in 1965 by Peter Collins [1].

As happens with living organisms, cities also evolve. Once upon a time this occurred very slowly, today they follow processes at an impressive speed that go hand in hand with the evolution of society, economy, and human behaviors. With cities, urban forms evolve and it becomes increasingly difficult to look at the changes in urban forms simply with the criteria of permanence or variation (if not permutation) of objects [2]. Other paradigms, linked to the resilience and adaptivity of populations and societies, are imposed today [3,4].

Due to the speed of transformation of urban spaces and objects, it increasingly appears interesting, not so much to record that an urban form has evolved from state A to state B, but rather to understand how that evolution occurred. In urban transformation systems with long inertia, such as pre-industrial urban systems (unless catastrophic events such as wars, earthquakes or fires intervened), they could easily be read by comparing the original configuration and the outcome. However, when faced with the speed of change

of the contemporary city today, we talk about transitional urban morphologies [5], we do not simply intend to grasp the general terms of the urban metamorphosis (the beginning and the end), but all the different phases that the urban form has gone through, perhaps capturing the possible deviations that the urban form did not take in the transition, favoring other paths.

In 1859, Charles Darwin dedicated an entire chapter of *On the Origin of Species* to the theme of morphological transition [6]. In his time, one of the surest tests of evolutionary theory consisted in creating taxonomies of fossil remains placed in chronological sequence, asking what the possible missing links were. However, in addition to working on these sequences of forms, Darwin began to imagine that there could be unexpected and fortuitous deviations in the tree of evolution. The sketch of the so-called "tree of life" (1837), found among Darwin's notes and now in the Museum of Natural Sciences in New York, in its imagining possible alternative paths to a linear structure of evolution, can constitute an extraordinary document for those who work on theories of urban design (Figure 1).

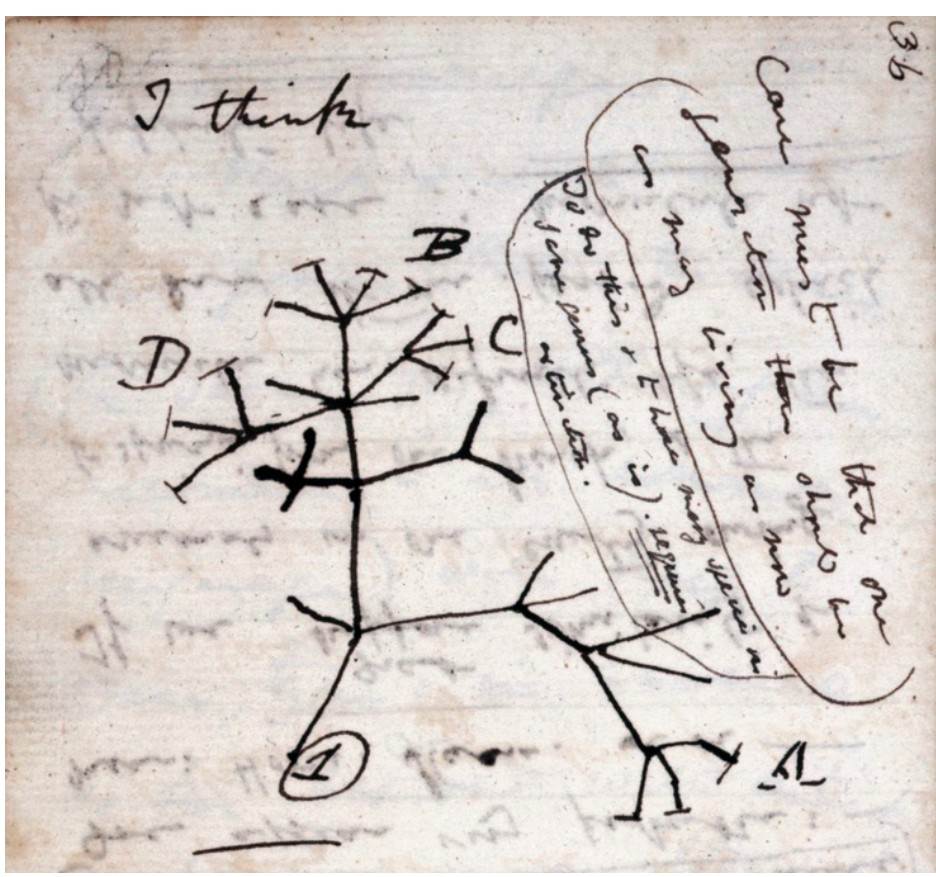

**Figure 1.** The sketch by Charles Darwin of the so called "Tree of Life" in his Notebook B "Transmutation of Species" (1837), p. 36. It shows the different paths of the national evolution, but also the different possibilities of evolution of a phenomenon in its own dynamic changing. (Public domain).

Due to the acceleration of multi-factor transformations affecting cities today, in the field of urban morphology analysis tools must be refined, not only technically, but also conceptually, perhaps even allowing us to re-read the shape of the cities of the past, which modern and post-modern urban morphology has probably frozen around an idea of a building type still anchored to its Enlightenment and rational origins (an idea entirely referring to the ideality of eighteenth-century European cities and the birth of the interrelationship between the form and function of architecture and spaces) [7].

The urban morphological discourse, which had an important propulsion from Italy between the 1950s and 1960s, capable of making its influence felt in Europe and in the world, has become blocked, at least in the last 25/30 years due to reasons of interpretative

rigidity and school orthodoxy [8]. This phenomenon has generated the crisis of urban design and of urban project, together with the crisis of theories of urban form. New slogans and new keywords have taken the place that the complex articulation of morphological discourse could have occupied.

The study of the dynamics of the dominant urban form in the work of Saverio Muratori and Gianfranco Caniggia [7], as well as the expansion of the method of research on the city to the modern geography advocated by Aldo Rossi [9], were first replaced by the rhetoric on the autonomy of architecture (compared to other disciplines) through the concept of urban type, then by the obsession with the permanent structure of objects and urban traces, finally by the evocation of an adaptive reuse of urban forms and architecture which, having arisen in the Anglo-Saxon context, seemed to adapt perfectly and simplistically to the rigidity of the architectural and urban types inherited from the past [10].

In the same years, between the end of the 1970s and the beginning of the 1980s, in the scientific context of the debate on natural evolution, the concept of adaptation entered into crisis due to the emergence of the concept of exaptation in the debate among evolutionary biologists [11]. This more recent idea, brought to the level of an epistemological paradigm, can still help urban morphology to escape from the impasse of interpretative rigidity and school orthodoxy. Exaptation can in fact, on the one hand allow us to recover attention for the forgotten transitional nature of urban structures that were already part of the investigations on urban form conducted by Muratori and Caniggia among others, on the other hand can allow us to rearrange the studies and the design of urban form for the new challenges that await it (Figure 2).

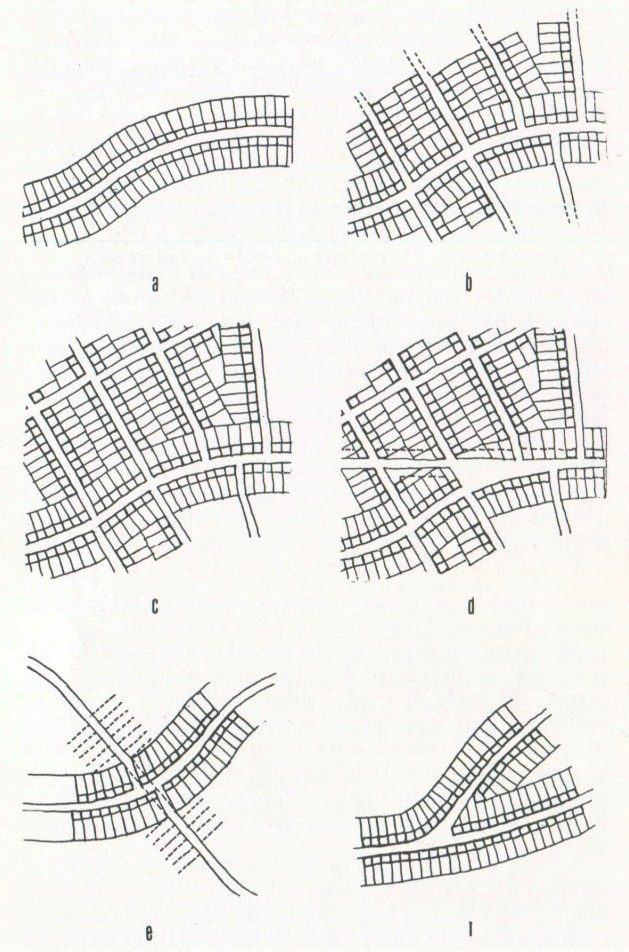

**Figure 2.** Scheme of formation and development of spontaneous systems in the studies by Gianfranco Caniggia (1972), published in Caniggia G. *Strutture dello spazio antropico. Studi e Note.* Alinea: Firenze, 1981, p.70. (Public domain).

They are challenges of substance and of methods. They are challenges of substance, because they manage the increasingly complex scenario of actors and factors capable of influencing the shape of contemporary cities. They are challenges of method, because they govern the framework of tools offered today by AI-based tools for the analysis and design of urban forms, between georeferenced multidimensional studies and parametric computational design [12,13].

In this general framework, the concept of exaptation, more and better than that of adaptation, can help to understand and manage the resilience phenomena that have become typical and evident in contemporary human settlements and which are destined to distinguish the cities of the future [14].

## 2. Methods

Urban morphology today, in the present conditions of human settlement, cannot ignore the identification, interpretation, and even criticism of what we could call here a theory of form, perhaps even (as Michel Foucault [15] would say) a social theory of form. The risk that, faced with the most urgent issues facing the world today (such as climate change, the reduction of resources within highly complex societies, progress in emigration/immigration), the urban form takes on a subordinate role, is a risk to be avoided.

The very scientific and philosophical foundations of morphological thought can come to the aid of a more detailed and refined consideration of the urban form on the one hand, and on the other, studies on the dynamics of transformation of the form can be given in timed intervals.

If, in regard to the scientific and philosophical foundations of morphological thought, the reference is undoubtedly Johann Wolfgang Goethe [16] with his intuition that there is a primordial form for each of the objects we study, Charles Darwin is crucial for those who want to read the primordial form in transitional terms, of continuous metamorphosis towards new and different structures [17], which still contemplate permanent features of the starting configurations and indeed, solicit reflections on the mechanisms of evolution themselves.

The reference to Goethe and Darwin (and to the seventy years that divide Goethe's studies on the metamorphosis of plants from Darwin's studies on the origin of species) serves here neither to establish a principle of authority nor to seek confirmation with respect to a general assonance of theories, but because it is instrumentally and conceptually necessary both in the field of analysis and in that of urban planning.

To study the phenomena of the transformation of reality, the theory of evolution constitutes an essential tool, not only in its conceptual bases and in its original development, but also in its subsequent outcomes, in the debates on the questions that remain open, and in the multiple articulations that arise they opened during the 20th century.

Stephen J. Gould in particular worked on the concept of exaptation in 1982 [11,18]. One of the crucial aspects of natural selection is the one of "adaptation", an already pre-Darwin concept, that describe the capability of an organism to fit to new functions. Adaptation became, in recent years, a widely used word in the architectural and urban designers' jargon; they often speak about adaptive architecture, adaptive spaces, and adaptive re-use [11]:

"Adaptation has been defined and recognized by two different criteria: historical genesis (features built by natural selection for their present role) and current utility (features now enhancing fitness no matter how they arose)" (p. 4, Abstract).

According to the first criteria, a feature is the result of a process aimed at satisfying a specific function: "a feature is an adaptation only if it was built by natural selection for the function, it now performs". According to the second criteria, a feature is something that augmented the capabilities of surviving because of its own peculiarity: in this case, adaption is "a static or immediate way as any features that enhances current fitness, regardless of its historical origin" [11].

In architectural design and in urban morphology we could evoke, for the first criteria, the interplay between form and function and, for the second criteria, the idea itself of building type, in its eighteenth-century meaning. In other words, architectural forms can

adapt following changes of function, or architectural forms can adapt because, in their being a fixed a determined typology, there is the capability in surviving.

Within a dynamic idea of urban morphology, by the point of view of transitional morphologies [5], we can enlarge the two ways towards adaptation from the interplay between forms and functions and from the idea of a fixed and non-transformable type, respectively to that process that we call permutation and the process that we call permanence.

Permutation is a concept coming from mathematics, describing each of several possible ways in which a set or number of things can be ordered or arranged. Generally speaking, it is a change that is built on what exists, just through its possible different configurations, for example in urban contexts the different uses of the same sequence of spaces. Permanence refers to, on the contrary, not to a change but to "what remains" due to its ability, given by the form, to resist time, for example hosting new functions without changing or changing only partially.

In the field of natural evolution, among Darwinian scientists, the existence of two (very) different criteria to read the adaptivity was not a matter of fact, but a real dilemma.

In the meanwhile, in 1966, Aldo Rossi published in Italian his book, The Architecture of the City, with its disciplinary critic to the "ingenuous functionalism", the American biologist George Williams published his book, *Adaptation and Natural Selection: A Critique of Some Current Evolutionary Thought* [19]. There, the author argues that "we must distinguish adaptations and their functions from fortuitous effects" [11], p. 5. Those fortuitous effect "always connotes a consequence following 'accidentally' and not arising directly from construction by natural selection" [11], p. 5. It was like opening a third way: in natural evolution, there are features coming neither from the adaptation to a function, nor from the adaptation of a form to new uses.

This dilemma was already before Darwin's eyes. Chapter 6 of the *On Origin of Species* (1859), entitled Difficulties on Theory, is the well-known and problematic chapter devoted to the "transitions". There, Darwin wrote about the strange case of the suture of young mammals' skulls:

"The sutures in the skulls of young mammals have been advanced as a beautiful adaptation for aiding parturition, and no doubt they facilitate, or may be indispensable for this act; but as sutures occur in the skulls of young birds and reptiles, which have only to escape from broken egg, we may infer that this structure has arisen from the laws of growth, and has been taken advantage of in the parturition of higher animals" [6].

This is the perfect representation of the adaptation dilemma. Sutures in the skulls of young mammals are not features built by natural selection for their present role of aiding parturition and furthermore they are not features ("no matter how they arose") enhancing fitness, because they are not specifically young mammals' features, but also of young birds and reptiles.

## 3. Materials

Permanence has traditionally been considered as a key concept for understanding the dynamics of transformation of urban form. Research has rightly focused on what remains, on objects that resist time over the generations, which simply continue to exist in the context of a transformative process. Controlling the permanence has for a long time been the only way to read the permutations or more generally the variations in the urban form.

In the last three decades, with the extension of the concept of resilience from classical physics to theories that consider human settlements as ecosystems [20], the idea of adaptation has become increasingly popular in the world of architectural and urban design, recalling the interest of scholars, professionals and decision makers [21]. Emblematic and now universally recognized is the definition of adaptation given in the architectural and urban field, and on the basis of the pioneering reflections of James Douglas [22], the ICOMOS New Zealand Charter for the Conservation of Places of Cultural Heritage Value in 2010:

"Adaptation means the process(es) of modifying a place for a compatible use while retaining its cultural heritage value. Adaptation processes include alteration and addition".

The transformation of urban objects and spaces over time can be easily described through the paradigm of adaptivity, developed within the framework of Darwinian evolutionary theories and now taken up within the architectural debate. However, upon closer inspection, the way urban forms behave in their transition from one structure to another (or to other various and different structures) seems to take on the characteristics of exaptation processes more precisely and more frequently. The fortuitous co-optation of existing forms seems to be the dominant criterion guiding the transition, much more than a simple adaptation of urban or architectural forms to new functions that alter or modify them.

Two case studies, briefly illustrated below, can help us understand the role of exaptation in urban transitions. These are the case of the medieval transformations of the Roman amphitheaters of the imperial age and the case of the transformations of the traditional Chinese courtyard house type of the Ming era during the second half of the twentieth century. Both cases have been often analyzed through the method of cultural-historical stratigraphy, today even pointed out as examples of urban recycling. Interpreting the same case studies from the point of view of exaptation can shed light on the very functioning of transitions that in the past have been interpreted with other criteria.

*3.1. The Morphological Transitions of the Roman Imperial Amphitheater between the Middle Ages and the Nineteenth Century*

The case of the Roman amphitheaters of the imperial age is exemplary. Aldo Rossi mentions them in the second chapter of his book, *The Architecture of the City* [9], *Primary Elements and the Concept of Area*:

"The Roman or Gallo-Roma cities of the West developed according to a continuous dynamic that exists in urban elements. This dynamic is still present today in their form. When at the end of the Pax Romana the cities marked their boundaries by erecting walls, they enclosed a smaller surface area than the Roman cities had. Monuments and even well populated areas were abandoned outside of these walls; the city enclosed only its nucleus. At Nimes the Visigoths transformed an amphitheater into a fortress, which became a little city of two thousand inhabitants: four gates corresponding to the four cardinal directions gave access to the city, and inside there were two churches. Subsequently, the city began to develop again around this monument. A similar phenomenon occurred in the city of Arles" [9], p. 87.

The "continuous dynamic that exists in urban elements" also concerns many other amphitheaters built in the Roman world in the first two centuries of the imperial age [23], each of them referable to the same type coded over time (Figure 3a).

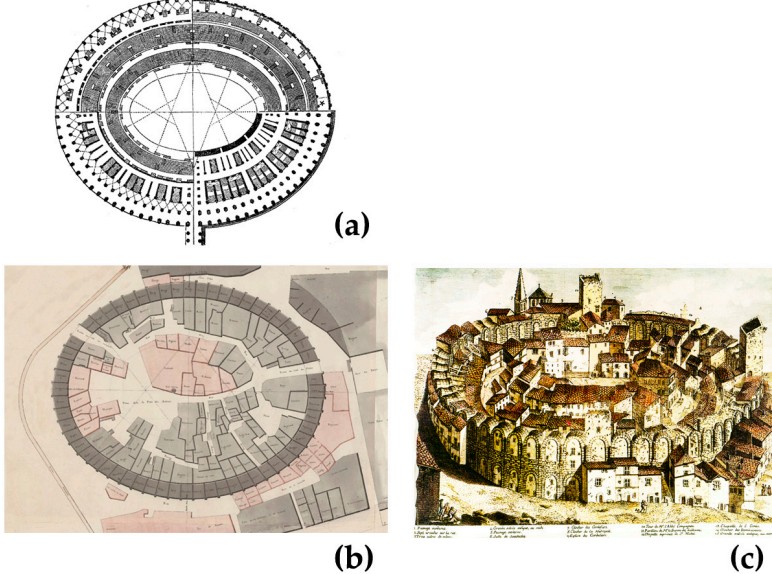

**(a)**

**(b)** **(c)**

**Figure 3.** *Cont.*

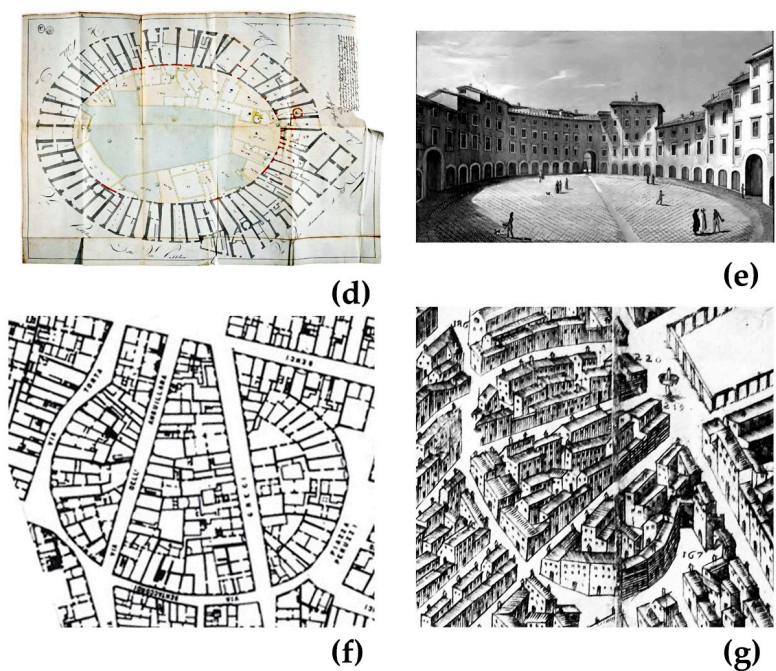

**Figure 3.** The three different "effects" from the same type (**a**) roman amphitheater (from Smith, W., *A Dictionary of Greek and Roman Antiquities*. London, 1875, s.v. "Amphitheatrum"): the fortresses of Nimes and Arles (**b**), "Plan de l'Amphithéâtre de Nimes" by S. V. Grangent 1809, (**c**), view of Arles Amphitheater by J. B. A. Guibert 1808), the square of Lucca (**d**), demonstrative plan of Amphitheater Square by Cardinali 1819, (**e**) view of the Market Square from Mazzarosa, A. *Guida di Lucca e dei luoghi più important del Ducato*. Lucca, 1843), the urban fabric of Florence close to Santa Croce (**f**), survey of the foundation walls 1970 ca. (**g**), Detail of the map of Florence by S. Bonsignori, 1584). Collected details from historical pictures by Author, 2023.

If self-contained fortified medieval villages were built on the amphitheaters of Nimes and Arles (Figure 3b,c, now disappeared due to nineteenth-century "cleanings"), in other situations the same urban element lends itself to different fates.

In Lucca (Figure 3d,e), the amphitheater gave rise to an elliptical square, also always maintaining a more or less large space in place of the arena in transition periods, before 1830–1834, in which the figure of the ellipse was still not so clear. In Florence (Figure 3f,g) the amphitheater has been absorbed by the urban fabric and even crossed by two streets such as Via dell'Anguillara and Borgo dei Greci, which, however, have not managed to completely erase the original imprint of the Roman structure, on the contrary the elliptical shape of the original type is still evident.

If we compare each of the three cases (Arles and Nimes together, Lucca, Florence) with the urban and building type of the amphitheater, often located peripherally in the Roman settlements of the imperial age, we realize the variety of solutions that, while remaining, the amphitheater type has allowed, even often involving the urban surroundings (as Rossi himself had suggested [9], p. 87). The round enveloping façades played the role of city wall precinct at Arles and Nimes, the empty space of the arena suggested the idea of a market square in Lucca, the structural system of the building type was co-opted to create a continuous urban fabric in Florence.

Thus, it may be interesting to read what remains in the transition from an amphitheater to a fortress, or to a square, or to a block and it may be equally interesting to ask ourselves the reasons for that type of permanence, or rather what rules the amphitheater followed in the evolve to something else and what paths it chose to vary. In the transition, the elliptical shape of the layout always remains evident, as do the radial walls substituting the *cavea* and, in section, the conical barrel vaults that are set on those radial walls.

In an era like ours, dominated by the pass-partout concept of adaptive reuse, the transformation of the Roman imperial amphitheaters speaks not of an adaptation to new functions of the ancient structure, but of a real co-optation of the monument by the inhabitants of each city, as the fortuitous effect of its intrinsic shape.

The radial layout of the walls, in fact, with the imposing barrel vaults, lend themselves to becoming part of a human settlements already organized on the prevalence of Gothic parcels. It is as if, unpredictably, each amphitheater concealed, in its own tectonics, a pattern capable of generating pieces of Gothic urban fabric to the point of forgetting the original project that had conceived it.

*3.2. The Morphological Transitions of the Chinese Ming and Qing Courtyard Houses in the Second Half of the Twentieth Century*

The case of the imperial Roman amphitheaters concerns the Western settlement culture of the classical world. However, cases of exaptation in the metamorphosis of urban forms and types can also be found in other geographical and cultural contexts, as well as in eras closer to our own. This aspect is all the more interesting the more it demonstrates the universality of the functioning of the transition processes from one urban form to another, a universality which can obviously then show its own specific characteristics in each specific geographical or historical reality.

Thus, an even clearer example of the interplay between permanencies and variations is that of the current fate of the urban fabrics made up of courtyard houses in the still surviving ancient parts of many Chinese cities [24,25].

In this case, the urban fabric, already very clear in its original constitution, disintegrated and then recomposed not by following plans of adaptation to new and different functions, but by evolving according to complexes processes dominated by fortuitous phenomena. Understanding these phenomena today is a challenge that is especially occupying scholars and designers involved in the regeneration of those urban areas, that is in understanding whether it is possible to trace in the processes of variation followed in the past methods and systems of transformation to be used in designing and planning for the future [26–29].

The Chinese courtyard house (*siheyuan*) has been organized since very ancient times through the development of a quadrangular courtyard overlooked by two single-story building blocks (maximum two floors): the access one and the actual living area. In the microcosm of the traditional courtyard house of the Ming era (1368–1664) and then of the Qing era (1644–1912), this system is replicated three times in line, giving rise to a sequence of three courtyards connected to each other via the residential blocks. The three courts correspond, in traditional matriarchal Chinese society, for example in Jiangsu (the province of Nanjing), to the three generations who share at the same time the management of the house, each with its own relevant court: the grandmother, the mother, the daughter.

After 1949, with the advent of collectivist society, that urban fabric based for centuries on a very clear family structure disintegrated. Each building block remained a living space, while each courtyard lost its role as the heart of family life to become a collective open space or part of public urban paths or even a storage space. Over time, this disintegrative process has also affected, in some cases, the harmonious differentiation between public and private uses (Figure 4).

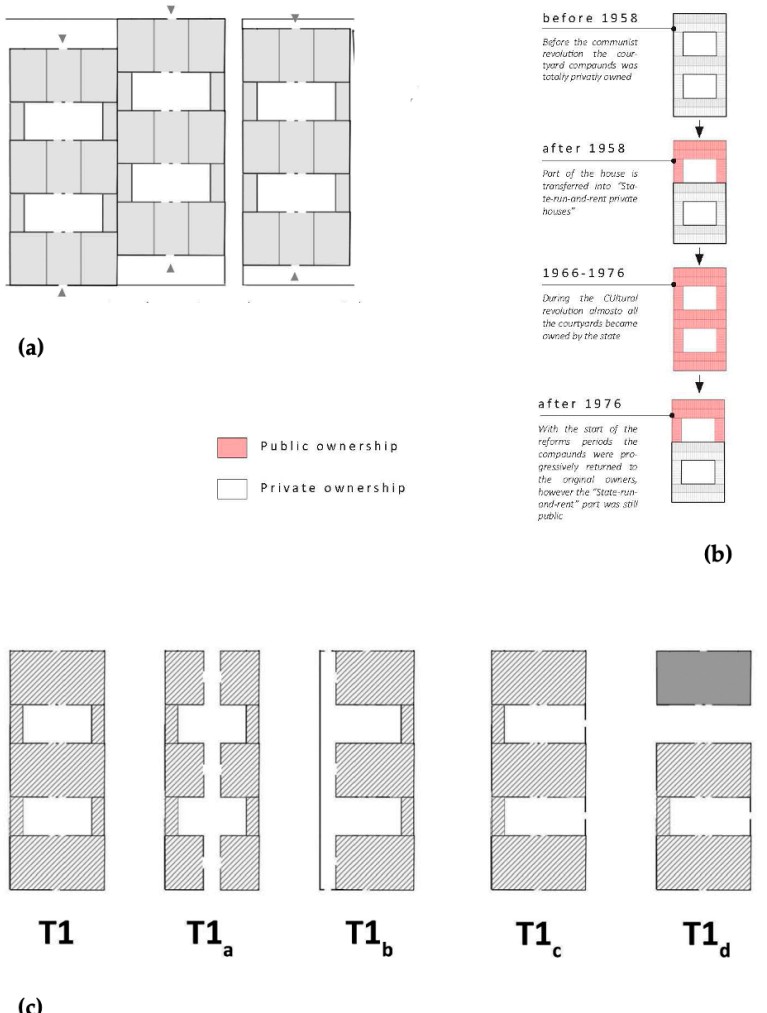

**Figure 4.** The disintegration/evolution of the Nanjing courtyard-house: from the recurrent of the same building types in the urban fabrics of Ming and Qing Dynasties (**a**), to the four more common new derived building typologies with different uses of the courts (**c**), because of the different uses (public-private or collective-individual) of the spaces (**b**), by Author 2018.

Today, in the oldest parts of some Chinese cities, large fragments urban fabrics from the Ming and Qing era can be found, originally based on the recurrence of the courtyard house type. They survived in front of the advance of other, more rational and functional urban building typologies in the 1970s and 1980s: workers' houses in line, four or five floors high and organized by sequences of stairwells.

However, those historic urban fabrics, after their disintegration have been subjected, especially in the last 30 years, to the phenomena of co-optation of their spaces for different activities. After the coming back of the property rights as the result of the economic reforms of 1990s in China, not only homes, but also small artisanal and commercial activities have occupied the covered spaces, while open spaces have begun to host vegetable gardens and small flower gardens, or courtyards for the expansion of domestic activities, as well as spaces for public relations for free time and also spaces for collective paths crossing the urban fabric. Sometimes, they have become places for building additions, extensions, excrescences, even very far from the natural development of the original types. In these transitional phenomena of the dynamics of urban forms, the type of courtyard house itself has demonstrated not only its adaptability to different uses, but a precise willingness to let its spaces and its surfaces be co-opted by societies, economies and uses that were very different from the original ones (Figure 5).

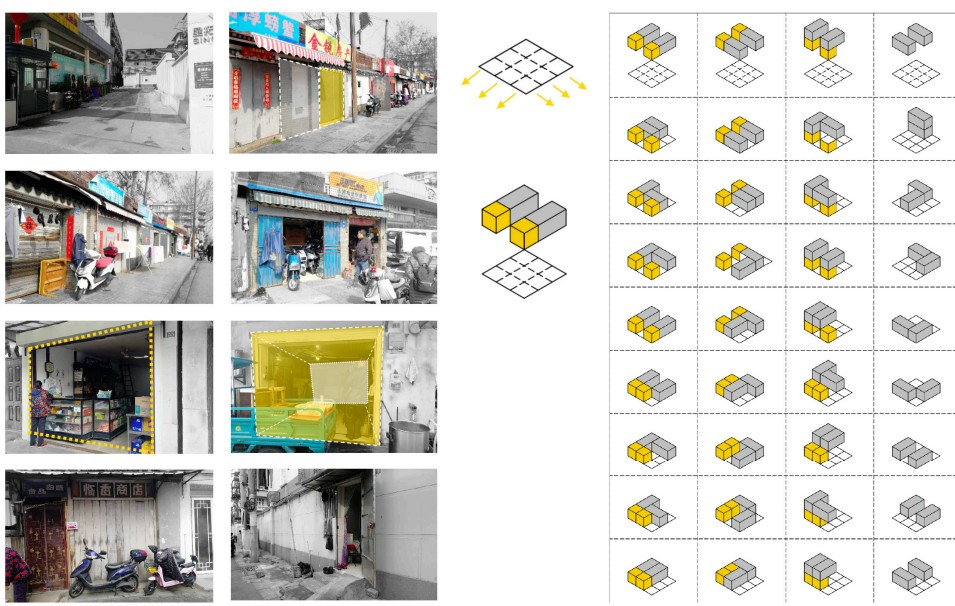

**Figure 5.** The co-optation of existing building types for new housing and micro-trading activities in Nanjing (Hehua Tang area): in yellow the new small shops, in grey residential spaces as legacy of the original courtyard-houses urban fabric. From the Design Studio "Urban morphology, architectural typology, contemporary settlement patterns" at SEUArch 2022 (Professors Bao Li and Marco Trisciuoglio).

That willingness was, obviously, also a willingness to metamorphose, to the point that the urban morphologist who deals today with the study of these fabrics struggles quite a bit to reconstruct the original structure. It must also be said that, if this type of reconstruction passes through almost archaeological practices, from the point of view of the urban regeneration project, a whole series of spontaneous solutions rise to extraordinary abacus of possible systems of new configurations [30].

Precisely as in the case of the Roman amphitheaters of the imperial age, also in the case of the courtyard houses of the Ming and Qing era in Nanjing, the shape of the original objects, or rather its intimate, tectonic and typological structure, is open to new and unexpected interpretations by who preordains its functions and uses. It is not about adaptive reuse, but about a real new life for the original objects which were not designed for that life or for similar destinations at all.

This means that in the horizon of the morphological transitions of urban spaces and objects there is not only the guiding criterion of resilience (adapting to new needs in order not to be demolished or removed), but there is something more complex and articulated such as availability of spaces and urban objects to new and unexpected transformations that make them evolve into new spaces and new urban objects.

## 4. Discussion

According to Darwin, there are phenomena escaping from the label of adaptation case. If the skulls of young mammals show sutures that are not the outcome of an adaptation process, it means that they are the effects of something different. George Williams again will nominate them "fortuitous effects" in 1966 [19]. Stephen J. Gould in 1982 will go deeper in the question, introducing the word "exaptation": ex-aptation versus ad-aptation [11].

In Gould's approach, there is the awareness of a significant lexical gap: the name to explain the matter is missing so that phenomena that could be relevant remain relegated to the peripheral areas of the evolutionary discipline. This lexical gap is part of a more general struggle, that concerns the structure of the disciplines as a structure for the construction, consolidation and development of knowledge as a place of discourse (in a decidedly Foucaultian sense).

Gould's approach can be summarized in a few sentences: "We suggest that such characters, evolved for other usages (or for no function at all), and later 'coopted' for their current role, be called exaptations (...). They are fit for their current role, hence *aptus*, but they were not designed for it, and are therefore not *ad-aptus*, or pushed towards fitness. They owe their fitness to features present for other reasons, and are therefore fit (aptus) by reason of (ex) their form, or *ex-aptus*. Mammalian sutures are an exaptation for parturition. Adaptations have functions: exaptations have effects. The general, static phenomenon of being fit should be called aptation, not adaptation" [11].

The conceptual paradigm shift proposed by Stephen J. Gould stays in considering that exists a set of aptations in anyone time and that it overlaps two subsets: "the subset of adaptations and the subset of exaptations). Gould describes this new look at the reality of evolution in a table, with the title "A taxonomy of fitness" (Table 1).

**Table 1.** A taxonomy of fitness (by Stephen Jay Gould, 1982, [11]).

| Process | Character | | Usage |
|---|---|---|---|
| Natural selection shapes character for a current use . ADAPTATION | AD-APTATION | APTATION | Function |
| A character, previously shaped by natural selection for a particular function (an adaptation), is coopted for a new use . COOPTATION | EX-APTATION | APTATION | Effect |
| A character whose origin cannot be ascribed to the direct action of natural selection (a non-aptation), is coopted for a current use . COOPTATION | EX-APTATION | APTATION | Effect |

In Gould's table, adaptation is function-based, while exaptations are effect-based. The theory of exaptation explained by Gould in 1982 immediately went beyond the boundaries of the studies on natural evolution, imposing itself (at least in Gould's mind) as a general theory of the world. From that moment, the exaptation theory took hold more as an interpretative paradigm than as a real scientific theory. The treatment that Gould reserves for the very writing of the essay, together with the South African scholar Elizabeth S. Vrba, has the character of a real manifesto to the detriment of the clearer development of a scientific essay.

This consideration of epistemological weakness is important here, since for the study of transitional urban morphologies the principle of exaptation certainly cannot assume a foundational and truthful meaning, but it has the meaning of proposing a point of view that opens up to new reflections and considerations. This was probably also Gould's original intent in the debate after and around the Darwinian theory of evolution; not so much to advance scientific theories by proposing a new truth capable of undermining the previous one, but to propose a doubtful point of view, to introduce a new interpretative concept, subjecting in a Popperian manner a consolidated theory now considered indisputable to the critical scrutiny of a reflection that manages to make it advance. Upon closer inspection, it is the same need that studies on urban form have demonstrated in recent years, blocked as they seem to be in reading reality once and for all without questioning too much about the dynamics of transformation of urban form and the factors that cause it.

A look at the many criticisms that have touched the concept of exaptation, as it was explained and exemplified by Stephen J. Gould, is useful for understanding, on the contrary, the meaning that small conceptual revolution had in the natural sciences and can also have in studies urban and in the general theory of urban form. In his monumental work and book-will, *The Structure of Evolutionary Theory* (2002), the Harvard paleontologist will use architecture to explain the concept of exaptation [31].

Chapter 11 is devoted to explaining the central role of exaptation in the macro-evolution, starting from a critic to the theory of adaptation. Gould refers to the spandrels of the domes of the Basilica di San Marco in Venice (it had already been the theme of an article published by him and Richard C. Lewontin in 1979, 3 years before the essay on exaptation) [32].

As Gould and Lewontin state, "Spandrels -the tapering triangular spaces formed by the intersection of two rounded arches at right angles- are necessary architectural byproducts of mounting a dome on rounded arches". They are there for geometrical and structural reasons: the dome, whose plan is a circle, could not insist on a square planned space without those spatial elements which allow the accordance between the circle and the square (Figure 6).

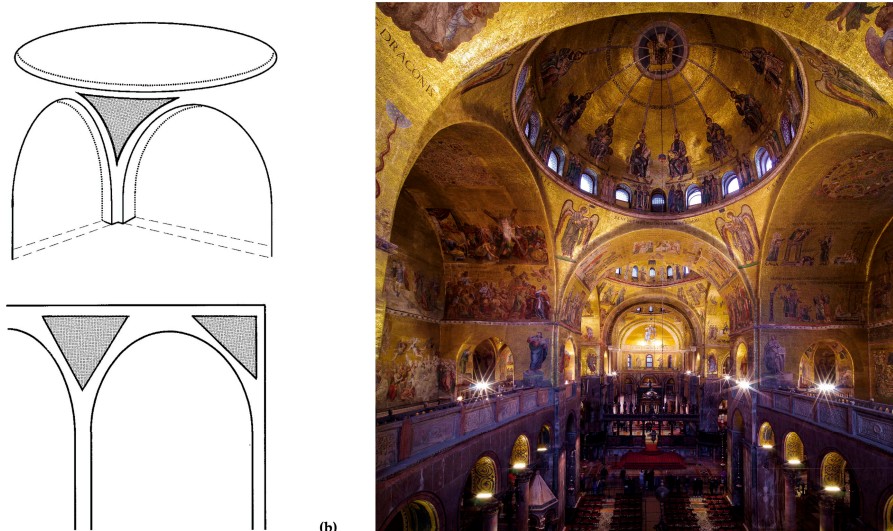

**(a)**                           **(b)**

**Figure 6.** "A pendentive (or three-dimensional spandrel) formed as a necessary triangular space where a round dome meets two rounded arches at right angles (upper). 'Classical' two-dimensional spandrels; the necessity triangular spaces between rounded arches and the rectangular frame of surrounding walls and ceilings (lower)", drawing re-drawn from [31] (**a**). The interior space of San Marco in Venice with the Byzantine vault and its spandrels, co-protagonist of the figurative program (picture by the Author) (**b**).

Spandrels are there for this reason and not to support the mosaic decoration of the Basilica. Nevertheless, within the context of the decorative program of San Marco's mosaics, the spandrels play a fundamental role, hosting each of them the image of an evangelist sitting in the upper part flanked by the heavenly cities and below a man representing one of the four biblical rivers (Tigris, Euphrates, Indus, and Nile) pouring water from a pitcher in the narrowing space below his feet. The comment by Gould and Lewontin is clear: "the system begins with an architectural constraint: the necessary four spandrels and their tapering triangular form", but "they provide a space in which the mosaicists worked; they set the quadripartite symmetry of the dome above" [32].

The reference to an element of architecture, rather than clarifying the concept of exaptation in its contrast to the concept of adaptation, makes this difference even more obscure. Gould's reflections were subjected, especially between the end of the nineties and the beginning of the new century, to harsh and insistent criticism (Gould died in 2002), both on the level of architecture and on the level of natural sciences. On the architectural level, in 1996, the architect and engineer Robert Mark, professor and researcher at Princeton, demonstrated the non-necessity of spandrels in the construction of a vault, since they "are design problems, not features that could either be designed or not" [33]. Mark offers a sort of explicit technical counterpoint to the criticisms of the Darwinian cognitivist philosopher Daniel C. Dennett, a strong advocate of the validity of the concept of "adaptionism" [17] for which the spandrels of San Marco "aren't spandrels (…). They are adaptations chosen from a set of equipossible alternatives for largely aesthetic reasons" [32].

These criticisms, to be honest, do not go unanswered. In 1997, Gould dedicated an entire article (The exaptive excellence of spandrels as a term and prototype) to counter each point, definitively contributing to making the term "spandrel" a metaphor/synonym of

the term exaptation. In that article, the Harvard biologist went so far as to publish his own drawing explaining the architectural meaning of the word spandrel, which would later become famous and would also be reproduced in other important works [30] (Figure 6a). There, above all, Gould clarifies the intent of his reflection of 15/20 years earlier:

"We wish (. . . ) to enrich evolutionary theory by a proper appreciation of the interaction between structural channeling (including the nonadaptive origin of spandrels as a central theme) and functional adaptation (as conventionally analyzed in studies of natural selection) for generating the totality and historically contingent complexity of organic form and behavior" [34], p. 10755.

Later, with a new generation of reflections, the criticism of adaptionism will become more retrospective and broader. Thus in 2000, the American philosopher and biologist Massimo Pigliucci recognized the existence in biology of an experimental component and a historical/philosophical component, explaining how Gould's positions should be recognized as being of great relevance in this second area [35]. Similarly, another American philosopher of science, Todd Grantham, will reconstruct in 2004 the fruitful and provocative role of Gould's writings at the same time, to be considered not as an opponent, but as a current supporter and continuer of Charles Darwin's evolutionary theories [36].

Anyway, according to the opinion of Gould and Lewontin, the decorative use of the spandrels, so relevant in the general economy of the decorative apparatus of the Basilica, is a case of fortuitous effect (as Williams could have said in 1966) or -better- it is a case of exaptation. Exactly as the sutures of the young mammals' skulls are not designed to allow parturition, the spandrels are obviously not designed to be decorated, but they are coopted to become the crucial support of the mosaic decoration of the Venetian Basilica. The same happened in the Middle Ages with the radial structures of the Roman amphitheaters of the imperial age or in the second half of the twentieth century with the courtyard houses of the Ming or Qing era in large Chinese cities such as Nanjing.

## 5. Results

In short, what appears evident in the case of the Roman amphitheaters and in the case of Chinese courtyard-houses urban fabric of the Ming and Qing Dynasties is that, alongside forms built to perform (adapting) to a new function and alongside forms built to survive over time thanks to their ability to adapt, there exist forms not designed to adapt to any function in particular, but who are available for new functions that chance entrusts to them, perhaps as needs to be fulfilled spontaneously (Table 2).

**Table 2.** A taxonomy of fitness in urban morphology (by Author, from Stephen J. Gould 1982, [11] Table 1).

| Process | Character | | Usage |
|---|---|---|---|
| URBAN FORM DYAMICS shapes character for a current use . ADAPTATION | AD-APTATION cases of PERMANENCIES and VARIATIONS (also PERMUTATIONS) of urban types of buildings and spaces, with the development of other urban types | APTATION | Function (affects the shape, imposing adaptations) |
| A character, **previously shaped by URBAN FORM DYNAMICS for a particular function (an adaptation)**, is coopted for a new use . COOPTATION | EX-APTATION CO-OPTATION of urban types of buildings and spaces, **originally developed for adaptation**, for new and unexpected forms of use (additions, extensions, excrescences) [*Chinese courtyard-houses*] | APTATION | Effect (by chance, and given by other factors, not necessarily by function) |
| A character, **whose origin cannot be ascribed to the direct action of URBAN FORM DYNAMICS (a non-aptation)**, is coopted for a current use . COOPTATION | EX-APTATION CO-OPTATION of urban types of buildings and spaces, **originally developed to perform a specific action**, for new and unexpected forms of use (additions, extensions, excrescences) [*Roman amphitheaters*] | APTATION | Effect (by chance, and given by other factors, not necessarily by function) |

As Gould himself states in the evolutionary context, the idea of exaptation does not replace that of adaptation, but broadens the framework of the general processes that produce the plurality of forms/features (in our case: urban forms/features), even opening to the idea of non-aptations:

"First, features may increase their representation actively by contributing to branching or persistence either as adaptations evolved by selection for their current function, or exaptations evolved by another route and coopted for their useful effect. Secondly, (...) features may increase their own representation for a host of nonaptive reasons, including casual correlation with features contributing to fitness, and fortuitous correlation found at such surprisingly high frequency" [11].

In the transfer to urban morphology, these words open to a wider understanding of the urban forms in the contemporary cities and even to a wider field of design solutions and a wider set of tools for developing urban projects, all based on the idea of flexibility and in the interplay between constraints and opportunities. Concluding his essay on exaptation, Gould writes:

"Flexibility lies in the pool o features available for cooptation (either as adaptations to something else that has ceased to be important in new selective regimes, as adaptations whose original function continues but which may be coopted for an additional role, or as non-aptations always potentially available). The paths of evolution -both the constraints and the opportunities- must be largely set by the size and nature of this pool of potential exaptations. Exaptive possibilities define the internal contribution that organism make to their own evolutionary future".

When he wrote the essay on exaptation, Gould not only makes a conceptual clarification on a point that was a flaw in the theory of evolution, but he raises the problem, evoking Michel Foucault, of the structure of knowledge in general. He knows very well that when we can understand why people classify in a certain way, we can also understand how people think and we know also that the taxonomies that follow one another over time are like the fossil traces of even substantial changes that have occurred in human culture.

What Gould claims for evolutionary morphology also applies to the discussion on urban morphology. Introducing the category of exaptation into the discourse on urban form today means suggesting a term that is missing in the taxonomy of urban morphology.

"Terms in themselves are trivial, but taxonomies revised for a different ordering of thought are not without interest. Taxonomies are not neutral (...); they reflect (or even create) different theories about the structure of the world".

## 6. Conclusions/Perspectives

Studies on the urban morphology of the contemporary city, dedicated to resolving issues of design of urban spaces and objects, cannot help but ask themselves the problem of the continuous metamorphosis of settlement forms. The transitional point of view allows the use of the most refined analytical tools (cultivated in the tradition of Italian morphological studies) on the living body of the contemporary Asian city (particularly the Chinese one), with recognizable settlement principles and multiple and varied transformation dynamics. The paradigm of urban transitional morphologies can lead to very interesting and innovative outcomes in new generation design practices, linked to parametric coding processes and machine learning hypotheses.

From the point of view of the epistemological approach of the problem of the transition from one (urban) form to another (urban) form, the comparison with the rich debate on post-Darwinian evolutionism can be of great relevance, also considering its ability to go beyond the limits of scientific disciplines and to constitute real general paradigms of looking at the world.

A new taxonomy of forms, proposed by Stephen J. Gould a few decades ago in the context of studies on the evolution of species, can find important applications in that part of studies on urban morphology which today, starting from instances relating to the renewal of the urban project as practice, questions the dynamics of transformation of the urban form as transitional dynamics, that is, interpretable by sequences of phases. In the debate

on the metamorphoses of spaces and objects in the city, the concept of exaptation allows us to escape from the dead end of adaptive reuse to think about the concept of permanence in terms of variability, flexibility, randomness (Figure 7).

This paradigm shift will allow urban morphology to free itself from the mechanical rigidity of Enlightenment/positivist typological schemes and to place itself in the best conditions to face the horizon of artificial intelligence.

If the parametric computational project, probably one of the most promising applications (at the moment) of artificial intelligence to urban morphology, allows us to work on multivariable models, exaptations can represent the key to bringing new taxonomies of urban forms to that type of design practice. Furthermore, some machine learning experiments dedicated to urban morphology problems (such as the relationship between block, street and building) demonstrate the effectiveness of the flexibility given by an adaptive approach in developing complex algorithms [37], also with the opportunity to establish fruitful connections between the collection of quantities of data, their spatialization in urban spatial forms and evidence-based design practices [38].

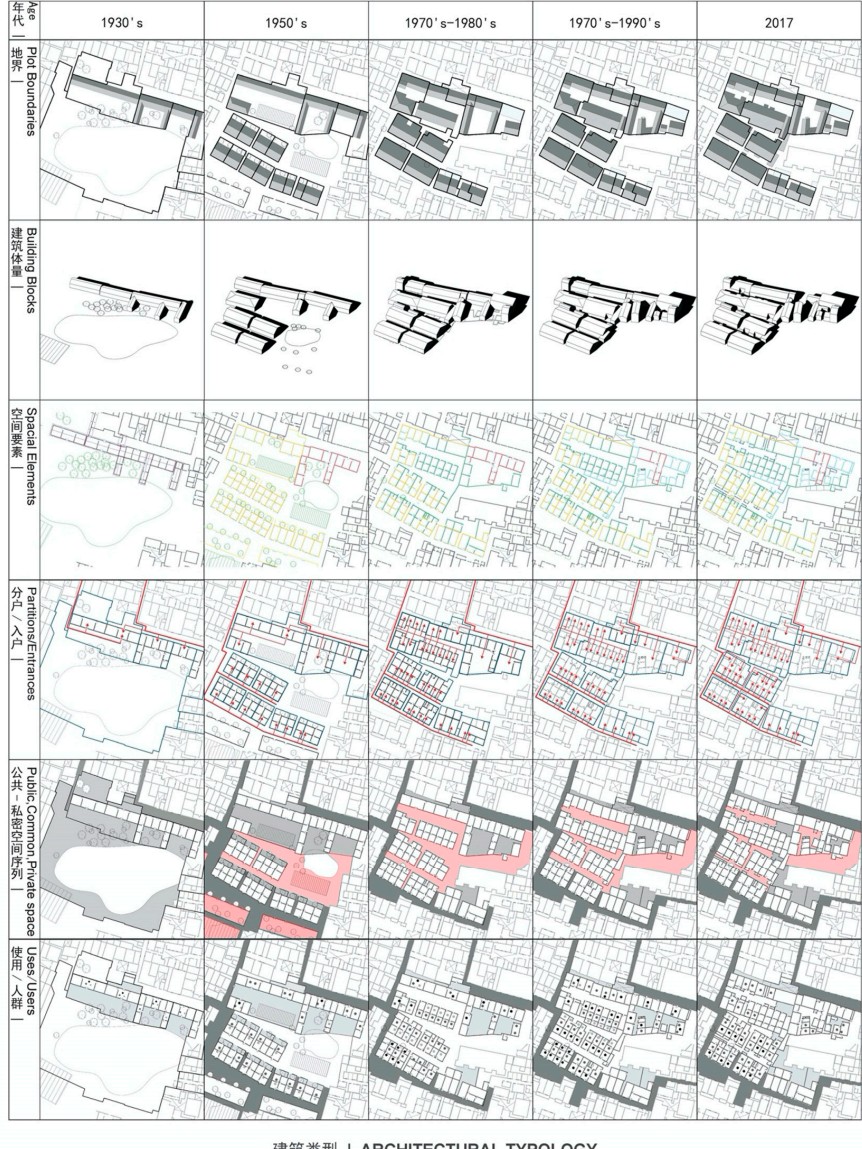

**Figure 7.** Studies on the transitional morphologies of the XiaoXiHu block in Nanjing (Qinhuai District), from the Design Studio "Urban morphology, architectural typology, contemporary settlement patterns" at SEUArch 2018 (Professors Bao Li and Marco Trisciuoglio).

Thus, since the road towards the applications of artificial intelligence to urban morphological analysis refers to machine learning practices, we can glimpse in the exaptations-based taxonomies an important pedagogical tool probably able to be extended to the project of new schools of urban design, re-evaluating precisely the role that innovative pedagogies can cover.

Furthermore, all this will finally allow a critical re-reading of the urban morphological discourse [39] from a morphogenetic and metamorphic point of view, opening the reflection on the urban form to themes of urban dynamics such as development and formal evolution, codes of control and promotion of changes, deformation, degeneration, epigenesis and heterogenesis, genealogy and reproduction, thus starting from the intuitions of D'Arcy Thompson [40]. It will be possible to grasp the importance of the role of topological studies and automatic recognition systems in urban analysis, and of the dialectic between evidence and intuition in the treatment of urban design problems. The time variable, which has never abandoned the dynamics of urban form [41], will thus finally find an adequate dimension of attention.

**Funding:** This research received no external funding.

**Data Availability Statement:** No new data were created or analyzed in this study. Data sharing is not applicable to this article.

**Acknowledgments:** The author acknowledges the support by the International Demonstration School Internationalization Demonstration School at Southeast University Nanjing and by the library of the "Transitional Morphologies" Joint Research Unit at Politecnico di Torino, in addition to the contribution of ideas, reflections and discussions received in recent years from the scholars (colleagues, researcher, PhD students, Master students) who animate the debate on urban morphology in both locations. The author expresses in particular his debt of gratitude towards his colleague Bao Li (Southeast University Nanjing, China), a companion in "transitional" studies and research between Italy and China, and towards Michela Barosio (Politecnico di Torino, Italy), with whom he discussed the contents of this work at length, receiving (once again) valuable suggestions.

**Conflicts of Interest:** The author declares no conflict of interest.

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
