# Peer review of "Exaptation in Transitional Urban Morphologies: First Notes on the Dynamics of Urban Form Read through the Theories of Natural Evolution"

_land, doi:10.3390/land13010074_

Round 1

Reviewer 1 Report

Comments and Suggestions for Authors

This document delves into the notion of exaptation within the context of urban morphology and the dynamics governing urban form. It expounds upon how preexisting structures, such as Roman amphitheaters and Chinese courtyard houses, have undergone co-optation and transformation to accommodate novel functionalities over time. The author posits that the concept of exaptation yields fresh perspectives on the evolving urban morphologies and their potential for pioneering urban transformation processes. Moreover, the document underscores the imperative for more sophisticated conceptual and technical instruments within urban morphology to align with the swift metamorphosis of urban spaces. In essence, the concept of exaptation broadens the comprehension of adaptation in urban morphology, enabling a more comprehensive consideration of permanence, variability, and adaptability in urban design.

In my opinion, it is worth publishing.

Author Response

Thanks for your kind review. I found your synthesis of my work in few lines really excellent and I decided to use some words of that to introduce the new version of my paper.

Reviewer 2 Report

Comments and Suggestions for Authors

It is generally known that a new term or definition will not help to solve a problem. On the other hand, we know that interdisciplinary transfers of interpretations of individual phenomena have an enriching effect and open up new segments of research. The introduction of the concept of exaptation into the interpretation of the urban organism also has this chance. The city was often compared and interpreted as a human or other living organism. Let's quickly mention just an example:

 The city as body

https://journals.openedition.org/articulo/1304

The stories in the article mentioned Roman amphitheaters are notorious. They were the subject of analyzes using the method of cultural-historical stratigraphy, or currently urban recycling. It would be appropriate to place the interpretation method of exaptation in a similar context. Let's leave general phenomenology aside so as not to get lost in the excessive breadth of the issue.

About the paragraph starting with line 81, the meaning of the presented research is very well summarized, it would be good to add how the exaptation method can help to build an idea of the resilience of the future city. It is currently a much-discussed challenge.

Line 97, this argument is fragile.

Line 148, here the author made a cultural-geographical leap, a short paragraph explaining that the exaptation method is not limited geographically or culturally would help.

Chapter 2.1 is more a case study of Nanjing courtyard-house than a part of general research.

3. Methods, ...the combination of Goethe, Darwin and Foucault in the space of a few lines slightly obscures the methodological intention of the author. This chapter repeats the statements from the previous parts of the text, the methods should have been described earlier.

Chapter 4. Discussion is based too much on Gould and does not have a solid framework. The argumentation is "fluid".

Even in Conclusions, the text acts as a review of Gould's work. The conclusion with figure 7 confirms the impression that it is a Nanjing Case Study wrapped in a broader theory.

Conclusion: the topic of the article has great potential. However, it is necessary to reorganize the text better both formally and content-wise. It must have a firmer methodological line. The theoretical background is blurry, it needs to get firmer outlines. It must be clear whether this is the incorporation of exaptation into the theory of architecture in general, as the title promises. Or whether it is an intercultural comparative study using the theory of exaptation.

General note: ALL figures must be identified by copyright, ...done by authors, or ...with permission to use intellectual property

Author Response

Really thanks for your detailed and relevant notes on my text.

I re-organize the content of the essay, above all exchanging paragraph 2 (materials, now 3) with paragraph 3 (methods, now 2), in order to inscribe the "materials" in a clearer methodological framework.

All the paragraphs have been also integrated in order to answer to your critics (and also the list of references has been extended).

Here I attach the new version of the text with highlighted added parts. Maybe it is the better way to let you realise my answers to each of your point.

Anyway:

  1. The introduction explains now the main idea of the paper (bringing exaptation in urban morphology studies to text it as paradigm), even citing the existence of a biological analogy in architectural discourse.
  2. The question of resilience is now quoted in paragraph 3 (from line 212) in order to better explain the debate between adaptation and exaptation in urban contexts.
  3. The first part of paragraph 3 has been extended in contents and references in order to let the arguments (supporting the choice of the two study cases) be stronger.
  4. The paragraph 3.1. (Roma amphitheaters) is now better showing the novelty of treating that typology and reading its metamorphosis in medieval time.
  5. The paragraph 3.2 (Chinese courtyard-houses) explains the choice of taking another study case far in space and more recent in time.
  6. There is now the attempt to treat the two study cases (3.1 and 3.2) in a similar way.
  7. The new version of paragraph 2 (Methods) put in order the roles of Foucault, Darwin and Goethe in the sequenze of quotation, with a wider argumentation.
  8. Paragraph 4 (Discussion) balances now the description of the theories by Gould with some of the more relevant critics he got (some from the world of architects), but also with the consideration that his thoughts of 1979/1982 were an important intellectual provocation more than a real scientific theory and such as that must be considered.
  9. At the beginning of paragraph 5 (Conclusion), I tried to highlight the general background and the general goal of this paper, coming from a research unit devoted to "transitional morphologies": the importance of treating the urban forms not as fixed objects of analysis, but as organisms in continuous development/dynamics.
  10. The credits of the pictures are described in captions. I am trying to have a confirmation about the public domain of Figure 1 and 2 (as it seems).

Thanks again for your helpful work and your consideration.

Reviewer 3 Report

Comments and Suggestions for Authors

Very interesting article and up to date studying the dynamics of urban form as a way of questioning the processes of evolution of the form in general. The authors explore Stephen Jay Gould’s ideas through the analyses of examples. The abstract is concise and provides sufficient information. The keywords are adequate. The introduction section presents a good framework which locates well the work and a good literature review to support it as well. The article describes very well the methodology and research methods. The results are consistent and represent a contribution to a better knowledge. The article is publishable, but the authors should, provide some minor corrections about some absent bibliographic references as it follows:

pp.4, lines 104-113 “p.8 236-238 The Roman or Gallo-Roma cities of the West developed according to a continuous dynamic that exists in urban elements. This dynamic is still present today in their form. When at the end of the Pax Romana the cities marked their boundaries by erecting walls, they enclosed a smaller surface area than the Roman cities had. Monuments and even well populated areas were abandoned outside of these walls; the city enclosed only its nucleus. At Nimes the Visigoths transformed an amphitheater into a fortress, which became a little city of two thousand inhabitants: four gates corresponding to the four cardinal directions gave access to the city, and inside there were two churches. Subsequently, the city began to develop again around this monument. A similar phenomenon occurred in the city of Arles”. which is the reference?

pp.8, lines 236-238 when the author cite: “Adaptation has been defined and recognized by two different criteria: historical genesis (features built by natural selection for their present role) and current utility (features now enhancing fitness no matter how they arose)” which is the reference? which is the reference?

pp.8, line 271 when appears this line with a citation: Those fortuitous effect “always connotes a consequence following ‘accidentally’ and not arising directly from construction by natural selection”. which is the reference?

Author Response

Thanks for your kind review and your suggestions.

In the new version of the paper I added quotes and I solve the questions of the three references you put me. Now they are at:

lines 246-255 (former 104-113), 150-152 (fomer 236-238), 184 (former 271)

Reviewer 4 Report

Comments and Suggestions for Authors

It would be interesting to have some more examples of exaptation in Urban Morphology in order to strengthen the methodological framework.

Author Response

Thanks for your kind comments.

After having considered a lot, I decided to write a new version of the paper where the methodological parts are better explained, also in a more extended way.

This is the reason why I kept only the two case studies of transformation in time (Roma amphitheaters and Chinese courtyard houses) not to let the paper become too heavy with other study cases.

The two chosen study cases can show the same phenomenon in two different places/cultures and in two different historical period. In my opinion this should be enough, in the economy of the paper, to illustrate the general questions.

Many thanks again to give me the opportunity to think more and more about the further possibilities of this study (I collected material that I will use for going deeper in showing examples in another occasion).

Round 2

Reviewer 2 Report

Comments and Suggestions for Authors

In terms of content, the article is sufficiently improved. 

I look forward to the response that the article will generate.